# LEARNING COMMUNICATION IN OFFLINE MULTI-AGENT REINFORCEMENT LEARNING

## ABSTRACT

Learning effective communication is one of the keys to improving coordination in multi-agent systems. This paper proposes a novel framework for offline multi-agent reinforcement learning that enables agents to learn effective communication from offline datasets that do not include any communication between agents. The proposed framework incorporates an attentional communication network into existing offline multi-agent reinforcement learning algorithms. Our experiments demonstrate the feasibility of learning effective communication from pre-existing datasets. In addition, we provide extensive analysis to examine how learned communication affects performance and to identify the characteristics of environments and datasets that enable effective communication learning.

## 1 INTRODUCTION

A key to advanced multi-agent systems, coordination is defined as the effective management of dependencies among agents (Malone & Crowston, 1990; 1994; Espinosa et al., 2004). One way to enhance coordination in multi-agent systems is introducing effective communication, which induces explicit dependencies among agents. Accordingly, in the field of multi-agent RL, learning effective communication has been actively studied, and it has been shown to improve performance in environments requiring coordination (Foerster et al., 2016; Sukhbaatar et al., 2016; Havrylov & Titov, 2017; Du et al., 2021; Sheng et al., 2022). Despite its effectiveness in terms of performance, learning communication often requires more data compared to when communication is not involved, which exacerbates the inherent problems of RL—sample inefficiency and the need for risky and expensive online interaction to generate samples.

Offline RL trains an agent from a fixed dataset previously collected by arbitrary algorithms, without requiring online environmental interaction. This is a promising way to address the aforementioned problems of RL and has thus been actively studied (Kumar et al., 2020; Fujimoto & Gu, 2021; Kostrikov et al., 2022; Ball et al., 2023). A primary challenge in offline RL arises from bootstrapping on actions absent from the dataset during policy evaluation, which introduces extrapolation errors. Since offline methods cannot generate new data, these errors accumulate over time (Kumar et al., 2020). This accumulation frequently results in the overestimation of value functions, hindering policy optimization. Recent works on offline RL have attempted to address this problem by constraining the target policy to be close to the behavior policy, which is assumed to have generated the dataset (Kumar et al., 2020; Fujimoto & Gu, 2021; Kostrikov et al., 2022). In offline multi-agent RL, this problem becomes even more severe with the increase in the number of agents due to the exponentially increasing action space Pan et al. (2022); Wang et al. (2024); Shao et al. (2024). Similar to single-agent algorithms, various approaches, including behavior-constrained methods (Wang et al., 2024; Kim & Sycara, 2025) and conservative value estimation techniques (Pan et al., 2022; Shao et al., 2024), have been proposed and demonstrated to be effective.

With the aforementioned limitation of communication and recent progress in offline multi-agent RL, one might naturally ask: "Can we learn effective communication from offline datasets?" Our focus is particularly on offline datasets generated by multiple non-communicating agents, a scenario frequently encountered in practice, to explore whether communication can emerge from datasets that have not accounted for interaction among agents. In addition, we explore how learning communication influences offline multi-agent RL algorithms and under what conditions effective communication, which enhances performance, can be learned from the perspectives of the environment and dataset.

In this paper, we propose a novel framework named `CODA`, which enables multiple agents to learn effective communication from offline datasets, particularly those generated by multiple non-communicating agents. As illustrated in Fig. 1, `CODA` aims to learn both policies and communication policies from offline datasets that do not include communication actions (i.e., messages) and then deploys them to environments allowing communication after the training ends. `CODA` involves an attention network for the communication policy and trains it by maximizing the cumulative reward through gradient flow from both other agents and itself. We build `CODA` on top of two behavior-constrained approaches in offline multi-agent RL algorithms, OMIGA (Wang et al., 2024) and FACMAC+B3C (Kim & Sycara, 2025), to evaluate its effectiveness and versatility.

We evaluate the performance of the proposed `CODA` across various multi-agent environments, including multi-agent Mujoco tasks and predator-prey scenarios, using datasets with differing degrees of partial observability and performance. The numerical results show that `CODA` outperforms SOTA offline multi-agent RL algorithms by successfully learning effective communication. In addition, we provide extensive analysis to investigate how `CODA` improves performance through learning communication and to identify the environmental and dataset conditions that enable effective communication, demonstrating, for example, that higher variation in returns within the dataset leads to performance gains from communication. Our contributions are summarized as follows:

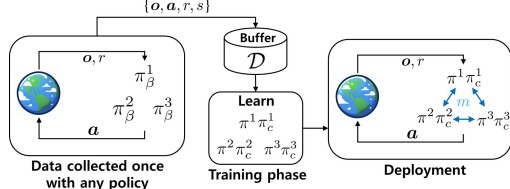

Figure 1: Learning both the multi-agent policy and communication policy from a given offline dataset generated by non-communicating agents. After training, the learned policies, $\pi^i$ and $\pi_c^i$, are deployed in the test phase. $m$ denotes the messages communicated between agents.

• To the best of our knowledge, this is the first study to introduce the problem of learning communication from offline datasets that lack communication data.

• Our approach can be integrated with any existing offline multi-agent RL algorithm and any communication strategy; we demonstrate this by using both OMIGA and FACMAC+B3C as subroutines.

• We provide an in-depth investigation of learning communication from offline datasets, considering algorithmic aspects, environmental characteristics, and dataset properties. We believe this offers meaningful insights into the newly introduced problem setup.

## 2 BACKGROUND AND RELATED WORKS

### 2.1 MULTI-AGENT RL

We consider fully cooperative multi-agent tasks, which can be modeled as a decentralized partially observable Markov decision process (Dec-POMDP) with $N$ agents (Oliehoek, 2012). The process unfolds as follows: at each time step $t$, the environment maintains a global state $s_t$, while each agent receives a local observation $o_t^i$, which represents part of the state. Based on this observation, each agent selects an action $a_t^i$ according to its policy $\pi^i$. The joint action $\boldsymbol{a}_t = (a_t^1, \cdots, a_t^N)$ determines the next state $s_{t+1}$, the subsequent observations, and a shared reward $r_t$. Through this interaction with the environment, the objective is to learn the optimal joint policy, $\boldsymbol{\pi}$ that maximizes the cumulative sum of team rewards, $\mathbb{E}\left[\sum_{l=0}^{\infty} \gamma^l r_l\right]$. To achieve this goal, recent multi-agent RL algorithms (Jeon et al., 2022; Kim et al., 2023) assume that the joint policy consists of independent policies, $\boldsymbol{\pi}(\boldsymbol{a}|\boldsymbol{\tau}) = \prod_{i=1}^{N} \pi^i(a^i|\tau^i)$, where $\tau^i$ is the local action-observation history of Agent $i$, enabling decentralized control at test time. These algorithms adopt the widely used centralized training with decentralized execution (CTDE) framework, where individual policies execute actions based on local information but are trained with the global state.

## 2.2 LEARNING COMMUNICATION IN MULTI-AGENT RL

To enhance coordination beyond the assumption of agent independence, multi-agent RL has explored learning communication, where agents exchange messages through a differentiable channel and incorporate them into decision-making. This allows agents to share partial observations (Foerster et al., 2016; Das et al., 2019) and intentions (Kim et al., 2021; Fang et al., 2023), leading to improved performance. A common approach is to train a communication policy via the channel to optimize other agents' policy and critic losses. For example, DIAL optimizes communication based on gradients of other agents' Q-function losses (Foerster et al., 2016), while intention sharing uses imagined trajectories as messages, extending beyond observation sharing (Kim et al., 2021). Other works study when (Jiang & Lu, 2018; Kim et al., 2019a) and whom to communicate with (Das et al., 2019), aiming for efficiency with low complexity (Kim et al., 2019b; Karten et al., 2023). Despite its success, communication learning remains largely unexplored in offline settings. In this paper, we investigate whether communication can be learned solely from offline datasets, how it affects offline learning, and under which conditions it is beneficial.

## 2.3 OFFLINE MULTI-AGENT RL

Offline RL aims to learn a policy that maximizes the expected return using a fixed dataset $\mathcal{D}$, which contains trajectories generated by arbitrary behavior policies, without requiring further environment interactions. Offline RL can be categorized into several approaches: behavior-constrained methods (Fujimoto & Gu, 2021; Xu et al., 2023), return-conditioned supervised learning (Chen et al., 2021; Kim et al., 2024a), value-based methods (Kumar et al., 2020; Kostrikov et al., 2022), and hybrid approaches that combine these techniques (Kim et al., 2024b). In this paper, we focus on behavior-constrained methods, which enforce policy regularization to keep the learned policy close to the behavior policy, thereby mitigating overestimation—one of the main challenges in offline RL—caused by selecting actions not present in the dataset. Two representative approaches in the single-agent setting include: Xu et al. (2023), which introduces a behavior regularizer by incorporating the divergence between the learning and behavior policies as an additional reward to discourage sampling unseen actions, and Fujimoto & Gu (2021), which applies a behavior cloning (BC) loss function as a regularizer to an existing online RL algorithm.

In multi-agent settings, overestimation becomes even more problematic as the action space expands with the number of agents (Yang et al., 2021). Moreover, in cooperative tasks, value factorization is often necessary to effectively assign credit for a shared reward by decomposing a centralized critic into individual critics. To tackle these challenges, numerous studies have been conducted (Jiang & Lu, 2021; Tseng et al., 2022; Meng et al., 2023; Zhu et al., 2023). We particularly focus on two extensions of the aforementioned behavior-constrained approaches to multi-agent settings. (a) One example is OMIGA (Wang et al., 2024), which incorporates the reverse KL divergence between the learning policy and the behavior policy into its objective, given by

$$\max_{\boldsymbol{\pi}} \mathbb{E}\left[\sum_{t=0}^{\infty} \gamma^t (r_t - \alpha f(\boldsymbol{\pi}(\boldsymbol{a}_t|\boldsymbol{o}_t), \boldsymbol{\pi}(\boldsymbol{a}_t|\boldsymbol{o}_t)))\right], \tag{1}$$

where $f(x, y) = \log(x/y)$ is the reverse KL divergence, and $\alpha$ is a hyperparameter controlling the degree of regularization. To optimize the objective, the joint Q-function and V-function are defined to satisfy $V_{jt}(\boldsymbol{o}) = \mathbb{E}[Q_{jt}(\boldsymbol{o}, \boldsymbol{a}) - \alpha \log(\boldsymbol{\pi}(\boldsymbol{a}|\boldsymbol{o})/\boldsymbol{\mu}(\boldsymbol{a}|\boldsymbol{o}))]$. In addition, both value functions are linearly decomposed into individual value functions are decomposed as follows: $Q_{jt}(\boldsymbol{o}, \boldsymbol{a}) = \sum_i w_i(\boldsymbol{o})Q_i(o^i, a^i) + b(\boldsymbol{o})$ and $V_{jt}(\boldsymbol{o}) = \sum_i w_i(\boldsymbol{o})V_i(o^i, a^i) + b(\boldsymbol{o})$, where $w_i(\boldsymbol{o})$ and $b(\boldsymbol{o})$ are the shared weight and bias functions respectively. Based on the decomposed value functions, the individual policies are optimized as follows:

$$\max_{\pi_i} \mathbb{E}\left[\exp(\frac{w_i(\boldsymbol{o})}{\alpha})\Big(Q_i(o^i, a^i) - V_i(o^i)\Big) \log \pi^i(a^i|o^i)\right] \tag{2}$$

(b) Another example is FACMAC+B3C (Kim & Sycara, 2025), which adds BC regularization directly to an existing online multi-agent algorithm, FACMAC (Peng et al., 2021), which employs a factored and centralized critic. In addition to BC regularization, FACMAC+B3C clips the target value of the Q-function in policy evaluation to the maximum return in the dataset to prevent the critic from diverging. The joint Q-function is updated to minimize the TD error with the critic clipping. The

corresponding objective function is given by, $\min_Q \mathbb{E}\left[\left(y^{jt} - Q_{jt}(s, \boldsymbol{o}, \boldsymbol{a})\right)^2\right]$, where

$$y^{jt} = r + \gamma \text{Min}\left[Q_{jt}(s', \boldsymbol{o'}, \boldsymbol{\pi}(\boldsymbol{o'})), R^*\right] \text{ and } R^* = M \max_d \sum_{t=1}^{T} r_{d,t}. \tag{3}$$

Here, $R^*$ represents the scaled maximum return from the given dataset, with $r_{d,t}$ denoting the reward at time step $t$ of the $d$-th episode. The joint policy is updated to maximize the cumulative reward and minimize the BC loss. The corresponding objective function is given by,

$$\mathbb{E}\left[-\boldsymbol{w}Q_{jt}(s, \boldsymbol{o}, \pi^1(o^1), \cdots, \pi^N(o^N)) + \beta \sum_{i=1}^{N} \underbrace{(\pi^i(o^i) - a^i)^2}_{BC}\right], \text{ where } \boldsymbol{w} = \frac{\alpha}{\frac{1}{N}\sum|Q_{jt}(s, \boldsymbol{o}, \boldsymbol{a})|}, \tag{4}$$

where $\alpha$ and $\beta$ are hyperparameters that control the balance between the RL objective and BC regularization, and Q-value is normalized to ensure robustness to scale. Note that a higher $\alpha$ places more emphasis on the RL objective while reducing the influence of BC regularization. This reduction, however, increases extrapolation error, causing instability in the learning process. For simplicity, we fix $\beta = 1$ in the paper. The key difference between OMIGA and FACMAC+B3C in terms of regularization is that OMIGA integrates regularization into the critic, thereby implicitly regularizing the policy through the critic, while FACMAC+B3C directly applies BC regularization to the policy.

Previous studies on offline multi-agent RL, including the two methods mentioned above, have effectively mitigated overestimation through various techniques. However, to the best of our knowledge, no prior work has explored learning communication from offline datasets. Thus, in this paper, we aim to introduce a method to enable multiple agents to learn communication and investigate the benefits of communication in offline multi-agent RL.

## 3 METHODOLOGY

### 3.1 SETTING

We introduce a new problem in offline multi-agent RL with communication. Specifically, given a multi-agent dataset that does not include any communication among agents, we aim to train both decentralized policies and communication policies that can be deployed in environments where communication is available, in order to maximize the expected return.

**Dataset.** The dataset $\mathcal{D}$ is assumed to consist of $D$ episodes: $\mathcal{D} = \{\boldsymbol{s}_d, \boldsymbol{o}_d, \boldsymbol{a}_d, \boldsymbol{r}_d\}_{d=1}^{D}$. Here, $\boldsymbol{s}_d$, $\boldsymbol{o}_d$, $\boldsymbol{a}_d$, and $\boldsymbol{r}_d$ are the sequences of states, joint observations, joint actions, and rewards in the $d$-th episode. For example, $\boldsymbol{o}_d = (\boldsymbol{o}_{d,t=1}, \cdots, \boldsymbol{o}_{d,t=T})$, where $\boldsymbol{o}_{d,t} = (o_{d,t}^{i=1}, \cdots, o_{d,t}^{i=N})$, and $o_{d,t}^i$ is Agent $i$'s the observation at time-step $t$ of $d$-th episode. Note that the dataset can be generated by any arbitrary policy or even by humans. In this paper, however, we use datasets generated by existing online multi-agent algorithms.

**Multi-agent RL with Communication.** Given the dataset, we train multiple agents, each with a policy and a communication policy, to maximize the cumulative sum of rewards, and then deploy them in environments where multiple agents can communicate with each other. We follow a scenario of communication built on top of the Dec-POMDP described in Sec. 2.1, which has been widely considered in the literature (Sukhbaatar et al., 2016; Singh et al., 2018). That is, at time step $t$, Agent $i$ observes an observation $o_t^i$ and communicate messages, $m_t = (m_t^1, \cdots, m_t^N)$, where $m_t^j$ is the message from Agent $j$. Here, we assume Agent $i$ has a policy $\pi^i(a_t^i | o_t^i, m_t)$ and a communication policy $\pi_c^i(m_t^i | o_t^i)$.

### 3.2 LEARNING COMMUNICATION FROM DATASETS

We now present a framework that learns to **C**ommunicate from **O**ffline **DA**tasets, named `CODA`. `CODA` enables multiple agents to improve coordination by learning effective communication from a given dataset without any interaction with either the environment or other agents. We incorporate `CODA` into two existing offline multi-agent RL methods, OMIGA and FACMAC+B3C, to explore communication learning in offline settings. This integration results in the proposed methods, OMIGA+`CODA` and FACMAC+B3C+`CODA`, respectively.

### 3.2.1 COMMUNICATION POLICY

In the integration of communication with both OMIGA and FACMAC+B3C, each decentralized actor has a policy and a communication policy. The communication policy is conditioned on its own observation, and the policy depends on both its own observation and the messages received from other agents. This connection between communication policies and other agents' policies enables gradient flow, allowing the communication policies to be trained through gradient updates. Building on previous research in multi-agent communication (Jiang & Lu, 2018; Niu et al., 2021), we introduce an attentional communication network that leverages multi-head attention (Vaswani et al., 2017) to design the communication policy. The architecture of this policy is outlined as follows: Each agent has its own encoder for both the policy and the communication policy, where $f^i(\cdot)$ is the fully connected layer for Agent $i$'s encoder, and the output of Agent $i$ is denoted as $e^i$. For the communication policy, each agent has an additional network, $f_c^i(\cdot)$. The outputs of all agents' encoders, $(f_c^i(e^i), \cdots, f_c^N(e^i))$, are fed into a self-attention module. Here, the self-attention module consists of key, query, and value components, and the output is the weighted sum of the value components, where the weight—the attention score—is determined by the dot product of the query and the key. The attention output is given by, $f_{\text{Attention}}(f_c^1(e^1), \cdots, f_c^N(e^N)) =$

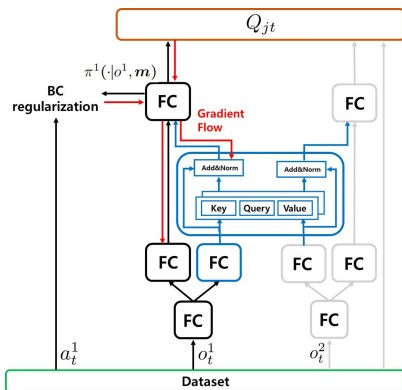

Figure 2: The policy and communication policy structure in FAC-MAC+B3C+CODA for the two agent case. Black boxes represent the policy, while blue boxes denote the communication policy. Both are trained with gradients derived from the RL objective and BC regularization.

$$\text{Softmax}(\frac{QK^T}{\sqrt{d_k}})V, \quad \text{where} \quad Q = \{W_q(f_c^i(e^i))\}_{i=1}^N, K = \{W_k(f_c^i(e^i))\}_{i=1}^N, \quad V = \{W_v(f_c^i(e^i))\}_{i=1}^N, \tag{5}$$

$d_k$ is the dimension of $f_c^i(\cdot)$, and $W_k$, $W_q$, and $W_v$ are the learnable parameters for the key, query, and value components, respectively. The output of the attention module then passes through a residual network and layer normalization before being fed into each agent's policy.

### 3.2.2 TRAINING

We train the policy and the critic in the same manner described in Sec. 2.3 for OMIGA+CODA and FACMAC+B3C+CODA. The communication policy is trained using the actor objective function, which aims to maximize cumulative rewards, following the prior works (Foerster et al., 2016; Sukhbaatar et al., 2016). The actor objective, integrated with the communication policy, is as follows: (1) The objective for OMIGA+CODA, is given by

$$\max_{\pi^i, \boldsymbol{\pi^c}} \mathbb{E}\Big[ \exp(\frac{w_i(\boldsymbol{o})}{\alpha})\Big(Q_i(o^i, a^i) - V_i(o^i)\Big) \log \pi^i(a^i|o^i, \boldsymbol{m})\Big] \tag{6}$$

where $\boldsymbol{m} = (\pi_c^1(o^1), \cdots, \pi_c^N(o^N))$ and $\boldsymbol{\pi^c} = (\pi_c^1, \cdots, \pi_c^N)$ is the joint communication policy and $\boldsymbol{m}$ is the message from all agents. Note that Eq. 6 is the objective function for agent i's policy and the communication policies. Thus, communication is trained through the gradients from both other agents and itself. Next, the objective function of FACMAC+B3C+CODA is derived by adding the communication policy to Eq. 4, as follows.

$$\max_{\boldsymbol{\pi}, \boldsymbol{\pi^c}} \mathbb{E}\Big[ \boldsymbol{w}Q_{jt}(s, \boldsymbol{o}, \pi^1(o^1, \boldsymbol{m}), \cdots, \pi^N(o^N, \boldsymbol{m})) - \beta \sum_{i=1}^N \underbrace{(\pi^i(o^i, \boldsymbol{m}) - a^i)^2}_{BC}\Big] \tag{7}$$

where $\boldsymbol{w} = \frac{\alpha}{\frac{1}{N}\sum |Q_{jt}(s, \boldsymbol{o}, \boldsymbol{a})|}$ and $\boldsymbol{m}$ is the same as defined in Eq. 6. We will investigate the feasibility of learning effective communication in both cases.

## 4 EXPERIMENTS

We conduct an in-depth investigation of the proposed method—learning effective communication from offline datasets—focusing on algorithmic aspects, environmental characteristics, and dataset characteristics. In this section, we aim to answer the following research questions.

**(RQ1)** Can offline multi-agent RL algorithms effectively learn communication from datasets generated by non-communicating agents?

**(RQ2)** What environmental characteristics facilitate effective communication learning among agents?

**(RQ3)** What impact does communication have on offline multi-agent RL algorithms?

**(RQ4)** What dataset characteristics enhance performance with communication?

### 4.1 EXPERIMENTAL SETUP

**Environments and Offline Dataset**    We evaluate our approach in three environments, each with various tasks: fully observable multi-agent MuJoCo, partially observable multi-agent MuJoCo, and a three-agent partially observable predator-prey scenario introduced in Peng et al. (2021), using various types of offline datasets.

• Three fully observable multi-agent MuJoCo environments: Multi-agent MuJoCo is a widely used benchmark for continuous-action robot control (Peng et al., 2021). It decomposes a robot into separate joints, treating each as an independent agent that should coordinate with others to accomplish tasks. In this setting, we evaluate three tasks: 3-Agent Hopper (HC), 2-Agent Ant (Ant), and 6-Agent HalfCheetah (HC). The offline datasets used in our tasks originate from OMIGA (Wang et al., 2024) and are generated by HAPPO (Kuba et al., 2021). We utilize four dataset types for each task: expert, containing high-performing trajectories from a near-optimal policy; medium, with trajectories from a moderately trained policy; medium-replay, comprising data from the stages of training; and medium-expert, which merges both medium and expert-level experiences.

• Two partially observable multi-agent MuJoCo environments: 6-Agent HalfCheetah (HC) and 5-Agent Swimmer (SW), each exhibiting varying levels of partial observability and performance. In both settings, an agent's observations are limited to its $K$ nearest neighbors, where $K$ controls the degree of partial observability. Specifically, we consider $K = 0, 1$ for HC, denoted as HC-k$K$, and $K = 0$ for SW. The offline datasets used in our tasks originate from FACMAC+B3C (Kim & Sycara, 2025) and are generated by ADER (Kim & Sung, 2023). For each task, we utilize six dataset types: expert, medium-1, medium-2, and their combinations.

• Partially Observable Continuous Predator-Prey Environment: This variant of the multi-agent particle environment incorporates partial observability, restricting each agent's perception. Unlike the original multi-agent particle environments (Lowe et al., 2017), agents can only detect the relative position and velocity of the prey within a predefined view radius, $Rad$. We evaluate this environment with $Rad = 0.5$, referring to it as PP-0.5. Here, we utilize three dataset types: expert, medium-1, and their combination.

In summary, we evaluate 27 cases across six distinct environments using different dataset types. Note that agent communication is not assumed during dataset generation.

**Baselines**    We compare the proposed framework, CODA, integrated with two offline multi-agent RL algorithms, and several baselines, including IQL (Yang et al., 2021), OMAR (Pan et al., 2022), and CFCQL (Shao et al., 2024), to mainly investigate if learning communication from datasets can improve performance. We report the performances of IQL, OMAR, CFCQL, and FACMAC+B3C from the results presented in Wang et al. (2024) and Kim & Sycara (2025), whereas we reproduce OMIGA and the proposed methods. We report the mean and standard deviation using three seeds.

**Hyperparameters**    For FACMAC+B3C, we have two hyperparameters: the RL coefficient $\alpha$ and the clipping value $M$. In addition, we have hyperparameters for the communication policy architecture, such as the dimension of the message. The detailed values we used is provided in Appendix A.

Table 1: Performance comparison of CODA and the considered baselines for a total of 27 cases across seven tasks and several datasets. Maximum and average returns are provided for each dataset. Dataset qualities are abbreviated as follows: 'expert' as 'e,' 'medium1' as 'm1,' and 'medium2' as 'm2.' Combinations of datasets are denoted as 'e-m1' for 'e' and 'm1,' and 'm1-m2' for 'm1' and 'm2.' Boldface numbers indicate the highest score or a comparable one among the algorithms. For brevity, we refer to FACMAC+B3C simply as B3C.

| Task-Dataset | IQL | OMAR | CFCQL | OMIGA | B3C | OMIGA+CODA | B3C+CODA |
|---|---|---|---|---|---|---|---|
| Fully observable multi-agent MuJoCo | | | | | | | |
| Hop-e | 755.7 | 2.4 | 718.5 | 1244.9±424.8 | **3619.7±1.6** | 1031.5±672.4 | 3592.8±25.1 |
| Hop-m | 501.8 | 21.3 | 674.2 | 975.5±294.4 | 3242.7±129.1 | 1091.2±51.6 | **3253.8±33.8** |
| Hop-mr | 195.4 | 3.3 | 1380.2 | 840.5±249.1 | 736.8±469.4 | **1622.0±441.8** | 1327.9±269.0 |
| Hop-me | 355.4 | 1.4 | 383.0 | 1313.4±647.2 | 3328.0±369.8 | 1430.6±817.5 | **3572.1±60.9** |
| Ant-e | 2050.0 | 313.5 | 1756.1 | 2048.6±11.8 | 2162.8±46.0 | 2052.0±6.2 | **2193.3±56.3** |
| Ant-m | 1412.4 | -1710.0 | 1159.6 | 1421.4±10.1 | **1516.5±14.8** | 1414.6±3.3 | 1521.5±838.1 |
| Ant-mr | 1016.7 | -2014.2 | 1052.9 | 1085.8±67.9 | **1259.8±302.4** | 1022.3±54.8 | 954.0±78.1 |
| Ant-me | 1590.2 | -2992.8 | 613.2 | 1291.8±869.0 | **2077.6±194.2** | 1894.7±44.1 | **2080.1±154.1** |
| HC-e | 2955.9 | -206.7 | 4999.2 | 3148.2±172.5 | **5403.5±169.7** | 3674.8±441.9 | **5447.2±453.3** |
| HC-m | 2549.3 | -265.7 | 4345.0 | 2684.2±43.8 | **4756.6±56.3** | 2544.0±85.8 | 4707.6±191.6 |
| HC-mr | 1922.4 | -235.4 | 3655.3 | 2339.4±205.6 | 4602.6±150.2 | 2469.0±164.0 | **4798.7±33.2** |
| HC-me | 2834.0 | -253.8 | 5030.9 | 2809.2±1021.1 | 5413.7±99.4 | 2622.5±77.6 | **5517.5±45.1** |
| Partial observable multi-agent MuJoCo | | | | | | | |
| HC-k0-e | - | 197.2 | 750.7 | **1390.1±5.2** | **1396.8±4.5** | 1388.1±3.6 | **1390.8±13.5** |
| HC-k0-m1 | - | 189.6 | 443.7 | 1106.3±1.1 | 1141.6±18.9 | 1119.0±10.0 | **1209.2±16.0** |
| HC-k0-m2 | - | 839.1 | 766.5 | 847.6±20.0 | 1195.0±51.9 | 839.0±16.2 | **1318.6±54.9** |
| HC-k0-e-m1 | - | 136.4 | 542.6 | 1199.5±14.1 | **1307.1±27.3** | 1129.9±38.6 | **1306.4±70.8** |
| HC-k0-e-m2 | - | 299.2 | 398.4 | 1097.9±19.3 | 1230.0±40.5 | 956.9±63.3 | **1330.0±24.0** |
| HC-k0-m1-m2 | - | 709.1 | 1186.1 | 1027.3±11.0 | 1210.1±22.8 | 1003.7±19.3 | **1395.0±13.1** |
| HC-k1-e | - | 3232.6 | 3390.2 | **3760.8±5.1** | **3760.5±24.2** | **3764.6±3.9** | 3786.9±41.9 |
| HC-k1-m1 | - | 2312.4 | 2356.0 | 2017.3±18.1 | 2508.1±55.7 | 1999.0±33.6 | **2645.0±53.8** |
| HC-k1-m2 | - | 1108.9 | 1440.6 | 1196.5±80.9 | 2187.8±66.7 | 1283.2±13.6 | **2338.2±53.8** |
| HC-k1-e-m1 | - | 2296.4 | 1949.6 | 2112.3±72.5 | 2682.0±175.4 | 2074.2±69.5 | **3762.1±64.1** |
| HC-k1-e-m2 | - | 524.8 | 864.7 | 1088.2±110.6 | 1102.8±349.8 | 1212.1±46.5 | **3320.6±319.2** |
| HC-k1-m1-m2 | - | 1410.6 | 1862.2 | 1633.5±48.0 | 2222.6±84.1 | 1525.3±158.0 | **2531.1±37.2** |
| Sw-e | - | 395.3 | 403.3 | **430.7±1.4** | **430.3±2.6** | **430.7±1.8** | 432.1±3.3 |
| Sw-m1 | - | 268.4 | 277.1 | **288.3±1.0** | **289.5±1.1** | **288.3±1.1** | 289.1±0.2 |
| Sw-m2 | - | 97.9 | 130.0 | 142.6±1.6 | **155.3±1.7** | 139.0±0.7 | 138.3±16.0 |
| Sw-e-m1 | - | 216.5 | 292.1 | 261.4±19.2 | 297.1±53.8 | 295.1±2.6 | **330.3±9.2** |
| Sw-e-m2 | - | 113.8 | 138.6 | 148.2±3.5 | 211.9±8.4 | 149.6±13.3 | **363.8±37.2** |
| Sw-m1-m2 | - | 226.4 | 222.7 | 205.8±20.7 | 189.9±7.7 | 209.7±36.8 | **276.6±3.1** |
| Partial observable predator-prey | | | | | | | |
| PP0.5-e | - | - | - | 186.1±61.9 | 183.5±4.5 | **192.4±52.0** | 189.1±56.6 |
| PP0.5-m | - | - | - | 40.9±9.2 | **87.7±29.1** | 43.1±13.4 | **88.6±32.3** |
| PP0.5-e-m | - | - | - | 91.7±28.0 | 163.4±83.5 | 129.7±24.1 | **183.6±56.2** |

## 4.2 OVERALL PERFORMANCE

The overall performances of the proposed method and the baselines are summarized in Table. 1.

**Fully observable multi-agent Mujoco.** Unlike partially observable environments, where communication benefits largely arise from sharing agents' observations, in fully observable environments, communication benefits can extend beyond observation sharing, such as sharing intentions, depending on whether tasks require intention sharing between agents. As seen in Table 1, FACMAC+B3C+CODA performs slightly better than FACMAC+B3C in some tasks, but in general, the communication gain is not large, indicating that sharing beyond observations is not necessary. However, for OMIGA, which underperforms compared to FACMAC+B3C, communication often improves performance. For example, in the Hopper environment with a medium-replay dataset, where both OMIGA and FACMAC+B3C struggle, OMIGA+CODA outperforms all considered baselines. In summary, for OMIGA, CODA improves performance in six of the twelve cases and reduces performance in two cases.

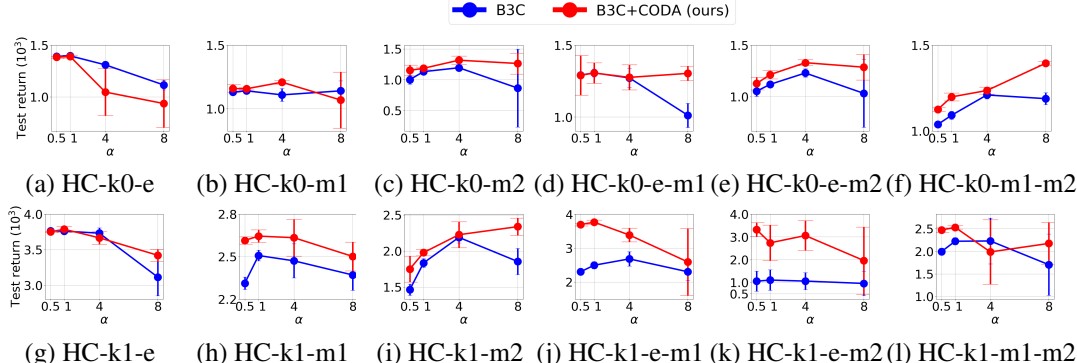

(a) HC-k0-e    (b) HC-k0-m1    (c) HC-k0-m2    (d) HC-k0-e-m1    (e) HC-k0-e-m2    (f) HC-k0-m1-m2

(g) HC-k1-e    (h) HC-k1-m1    (i) HC-k1-m2    (j) HC-k1-e-m1    (k) HC-k1-e-m2    (l) HC-k1-m1-m2

Figure 3: Performance comparison of FACMAC+B3C and FACMAC+B3C+CODA across different values of the RL coefficient $\alpha$ in the Halfcheetah environment. A higher $\alpha$ indicates that less BC regularization is applied.

**Partially observable environments.** In the partially observable environments, multi-agent Mujoco and the predator-prey scenario, FACMAC+B3C outperforms B3C and other baselines, with significant margins in certain cases. The communication gain is particularly notable in HC-k1-e-m1, HC-k1-e-m2, and Sw-e-m2, where FACMAC+B3C without communication fails to match the average return from the dataset. In these environments, CODA achieves the best performance among the considered algorithms. In contrast, OMIGA struggles to leverage communication gains in partially observable settings and, in many cases, performs even worse.

The overall results show that either FACMAC+B3C+CODA or OMIGA+CODA outperformed the baselines in 31 out of 33 cases across 7 environments with various datasets, which answers **RQ1**.

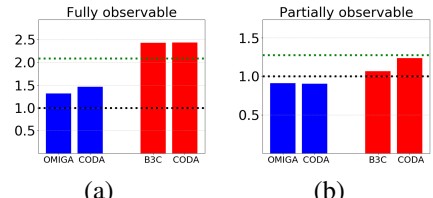

(a)             (b)

Figure 4: Normalized performance comparison of OMIGA (blue), OMIGA+CODA (blue), FACMAC+B3C (red), and FACMAC+B3C+CODA (red) in (a) fully observable environments and (b) partially observable environments. The black dotted line and the green dotted line denote the average return and the maximum return of the datasets, respectively.

### 4.3 ANALYSIS

In this subsection, we analyze how CODA enhances performance through effective agent communication, and how environment and dataset characteristics—such as partial observability and return variance—affect communication learning in offline settings.

#### 4.3.1 PERFORMANCE GAIN FROM COMMUNICATION

In this section, we investigate the performance gains achieved by incorporating communication into OMIGA and FACMAC+B3C individually. We further analyze the communication gains in both fully observable and partially observable environments. Fig. 4 illustrates the performance of OMIGA, FACMAC+B3C, and their CODA-integrated versions in these environments. The performance is averaged across tasks and datasets, with each value normalized by the mean return of the corresponding dataset. In fully observable environments, FACMAC+B3C already surpasses the maximum return of the dataset, indicated by the green dotted line, demonstrating that incorporating communication does not yield further improvements. However, CODA is shown to enhance the performance of OMIGA. On the other hand, in partially observable environments, FACMAC+B3C benefits from communication with CODA, while OMIGA does not and performs worse than the dataset quality. It is observed that FACMAC+B3C+CODA nearly achieves the maximum return in partially observable environments. Summarized above, we observe that communication offers a gain particularly to FACMAC+B3C in partially observable environments but not in fully observable environments. However, a performance gain is achieved in fully observable environments for underperforming algorithms. These observations answer **RQ2**.

### 4.3.2 Impact of Communication on Offline Learning

In multi-agent RL, communication is known to improve performance (Kim et al., 2021). As shown in the previous section, the performance gain from communication in partially observable environments is greater than in fully observable environments, particularly for FACMAC+B3C. Here, we offer a different perspective on how communication enhances performance within the context of the behavior-constrained approach. Fig. 3 shows the performances with and without `CODA`, by varying the RL coefficient $\alpha$ in Eq. 7. As discussed in Sec. 2.3, the BC regularization reduces extrapolation error by encouraging the learning policy to remain close to the behavior policy. As $\alpha$ in Eq. 7 increases, less BC regularization is applied, resulting in more severe extrapolation. Thus, in Fig. 3, it is clearly seen that $\alpha = 8$ yields worse performance than $\alpha = 4$ in FACMAC+B3C in all cases. This indicates that the extrapolation error becomes severe when $\alpha > 4$. However, FACMAC+B3C+`CODA` is more robust to higher $\alpha$ values, and even $\alpha = 8$ yields the best performance in some cases, such as HC-K0-m1-m2 and HC-k1-m2. This implies that `CODA` requires less BC regularization by alleviating extrapolation through communication. The above results answer **RQ3**.

### 4.3.3 Communication Gain and Data Characteristics

We investigate the conditions under which learning communication improves performance, focusing on dataset characteristics. In Sec. 4.3.1, we show that, for FACMAC+B3C, communication learning enhances performance in partially observable environments. Here, we further explore how dataset characteristics affect this performance gain. Our analysis reveals that the return distribution, particularly its variance, influences the effectiveness of communication learning. To measure this, we calculate the coefficient of variation (CV) Abdi (2010), which indicates the relative spread of returns in the dataset. The CV is defined as the ratio of the standard deviation to the mean. We compute the CVs for the partially observable Mujoco datasets and analyze the relationship between the CV and the performance gain from learning communication. Fig. 5 shows the performance gain with respect to the CVs in partially observable MuJoCo environments, where each circle represents a specific dataset. It is observed that the performance gain increases as the CV increases. This implies that more diverse datasets, in terms of episode quality, enable RL agents to learn effective communication. The revealed relationship between the CV of data returns and performance gain from learning communication answers **RQ4**.

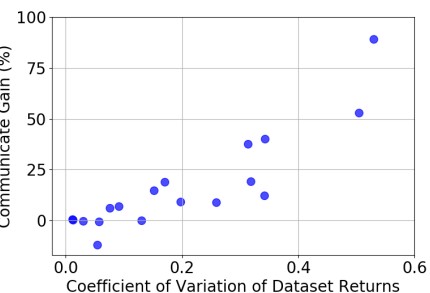

Figure 5: Performance gain from learning communication with respect to the coefficient of variation of dataset returns in partially observable MuJoCo environments. Each circle represents the performance gain for a specific MuJoCo environment with a particular dataset. For example, the circle in the upper-right shows the normalized performance gain of FACMAC+B3C+`CODA` over FACMAC+B3C in HC-k1-e-m2.

## 5 Conclusion

We propose a novel framework, `CODA`, which enables multiple agents to learn effective communication from offline datasets generated by arbitrary non-communicating agents. By integrating `CODA` with two existing offline multi-agent RL methods, we investigate the feasibility of learning communication, its impact on offline learning performance, and how key dataset and environment characteristics influence the performance gains achieved through communication learning. Our numerical results and detailed analyses demonstrate that incorporating communication can significantly improve performance under various conditions, particularly in partially observable environments.

**Limitation and Future work.** While our method empirically shows the feasibility of learning communication offline, it lacks a theoretical foundation. Important aspects such as when, what, and whom to communicate remain open for future investigation. In conclusion, we believe that the proposed framework, `CODA`, will inspire further research and drive new advancements in offline multi-agent RL, specifically in the area of learning communication strategies from limited data.

**Ethics Statement.** We affirm adherence to the ICLR Code of Ethics. This work does not involve human subjects research and personally identifiable information. All experiments are conducted in simulated environments and/or with publicly available datasets used under their respective licenses. We are not aware of privacy, safety, or discrimination risks specific to our setting, and we identify no conflicts of interest.

**Reproducibility Statement.** Our implementation builds upon publicly available open-source baselines that we properly cite. To ensure reproducibility, we provide detailed descriptions of the datasets, preprocessing procedures, model architectures, and hyperparameters in the Appendix. The main paper describes the evaluation protocols. These resources together allow independent researchers to replicate our results without access to the implementation itself.

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

# A HYPERPARAMETERS

We conducted the experiments on a server with dual AMD EPYC 7713 CPUs and an NVIDIA RTX 6000 Ada Generation GPU. Each experiement took about 6 hours.

We here provide the hyperparameters of CODA. We have three hyperparameters: the RL coefficient $\alpha$, the clipping value $M$, hidden dimension of the communication vector $d_k$. We use $d_k = 8$ for all considered environments. In addition, We use the same value for $M$ with FACMAC+B3C. Thus, we set $M = 1$ except for one task (Ant, medium-replay dataset). The values for $\alpha$ and $M$ are provided in Table. 2.

Table 2: Hyperparameters for multi-agent Mujoco environments

| Environment | Dataset | $\alpha$ | $M$ |
|---|---|---|---|
| Hopper | expert | 16 | 1 |
| | medium | 8 | 1 |
| | medium-replay | 0.25 | 0.1 |
| | medium-expert | 1 | 1 |
| Ant | expert | 1 | 1 |
| | medium | 1 | 1 |
| | medium-replay | 1 | 1 |
| | medium-expert | 0.5 | 1 |
| HalfCheetah | expert | 16 | 1 |
| | medium | 16 | 1 |
| | medium-replay | 16 | 1 |
| | medium-expert | 16 | 1 |
| HC_obsk0 | expert | 1 | 1 |
| | medium1 | 1 | 1 |
| | medium2 | 8 | 1 |
| | expert-medium1 | 1 | 1 |
| | expert-medium2 | 0.5 | 1 |
| | medium1-medium2 | 1 | 1 |
| HC_obsk1 | expert | 1 | 1 |
| | medium1 | 4 | 1 |
| | medium2 | 4 | 1 |
| | expert-medium1 | 1 | 1 |
| | expert-medium2 | 4 | 1 |
| | medium1-medium2 | 8 | 1 |
| Swimmer | expert | 1 | 1 |
| | medium1 | 1 | 1 |
| | medium2 | 1 | 1 |
| | expert-medium1 | 4 | 1 |
| | expert-medium2 | 1 | 1 |
| | medium1-medium2 | 0.5 | 1 |
| Predator-prey | expert | 8 | 1 |
| | medium1 | 8 | 1 |
| | expert-medium1 | 8 | 1 |

