# OpenReview forum: "Learning Communication in Offline Multi-Agent Reinforcement Learning"
_ICLR.cc/2026/Conference — ICLR 2026 Conference Withdrawn Submission_

### Official Review · Reviewer_PSpP · 2025-10-30

**Soundness:** 2
**Presentation:** 3
**Contribution:** 2
**Rating:** 4
**Confidence:** 4

**Summary:**

This paper introduces **CODA**, which enables multiple agents to learn communication from offline datasets generated by non-communicating agents. CODA is integrated into two offline algorithms and evaluated on two multi-agent environments with different observability settings. The results show that, with CODA, offline MARL methods can achieve better performance in some tasks.

**Strengths:**

Overall, the paper is easy to follow, and the idea of learning communication from non-communicating offline data is interesting.

**Weaknesses:**

Combining communication learning with offline MARL is interesting. Nevertheless, the communication learning techniques in this paper largely follow existing literature, and it remains unclear whether any adaptations are necessary for the offline MARL context, given the challenges in the offline setting. Moreover, the paper lacks a deeper investigation into how communication learning influences offline MARL, particularly with respect to the learning objectives of the underlying algorithms.

**Questions:**

- I am confused about the discussion of recent progress in offline multi-agent RL (especially in the second paragraph of the Introduction). I do not see a clear connection between learning communication and the challenges in offline RL. Would learning communication help address these challenges?

- The related work section discusses the overestimation problem. Could learning communication further exacerbate this issue?

- Line 207: The cited works, _CommNet_ and _IC3Net_, do not explicitly include a communication policy. Instead, communication is integrated into the policy network. Please provide the correct citations.

- The paper claims to use multi-head attention, but Equation (5) corresponds to Scaled Dot-Product Attention. Could you provide the correct formula or clarify the implementation?

- The results are evaluated using only three random seeds. It is unclear how the results in Table 1 were computed. In addition, given the potentially high variance of the methods, performance curves should be provided to illustrate the learning process.

- Line 337: Please explain why CODA reduces performance in two cases.

- Some results appear statistically insignificant due to high variance (e.g., _Ant-m_, _Ant-me_, _HC-e_, _HC-m_, _HC-k0-e_, _HC-k0-e-m1_, _HC-k1-e_, _Sw-e_, _Sw-m1_, _PP0.5-m_). Please provide statistical tests to confirm the significance of these results.

- Section 4.3.2 is somewhat confusing. It analyzes the impact of the BC and RL objectives, but it is unclear how communication learning affects these two objectives. Could you elaborate on this interaction?

---

### Official Review · Reviewer_ZSxn · 2025-10-31

**Soundness:** 3
**Presentation:** 2
**Contribution:** 1
**Rating:** 2
**Confidence:** 4

**Summary:**

This paper presents a framework for communication in offline multi-agent reinforcement learning, i.e., the messages are learned from trajectories that are previously collected without any implicit communication. The proposed method shows to be able to learn communication from previous datasets where communication was not present.

**Strengths:**

- this paper provides an extensive set of experiments; the environments used are tested under several different configuration which gives a good experimental breadth to this work
- communication in MARL is a relevant topic, and it is interesting to see it being applied in offline MARL

**Weaknesses:**

While this paper tackles a relevant and interesting problem in MARL, there are still several points that still need improvement.

Firstly, there are also other online MARL cases that can be solved both with and without communication, meaning that in some sense we would still be learning communication from an environment that does not have implicit communication; i.e., it doesnt need to be an offline dataset for agents to learn communication in tasks where agents could also be non communicating agents instead (as mentioned in lines 55-56).

When being tested in the actual environments (after the offline training), it would be interesting to see if the policies learned by the proposed method achieved performances that are comparable to other online methods; it would also be interesting how the proposed method performs and compares with other methods such as QPLEX [3], which can act both online and offline.

In section 4.2 the authors provide experiments in fully observable settings; it could be argued that communication in fully observable environments might be redundant and just overhead for the learning process of the agents since they can observe the others at any time and do not require any information to be shared by them; it is however a good point raised in line 370 by the authors that intentions can be shared, and hence, in these cases, it would be helpful to know better what the agents are communicating with each other; it is not entirelly clear at the moment.

Overall, i believe the novelty of this work is limitted, since it seems the only difference from other works is that here a communicaiton attention-based architecture is applied on offline MARL, which is something that has been extensivelly explored in online MARL. Please find below some more specific questions.

**Questions:**

1. following from some of the previous points: "to the best of our knowledge, this is the first study to introduce the problem of learning communication from offline datasets that lack communication data" (lines 80-81) - how is this different from learning communication in an online marl task that does not have implicit communication?

2. how is the proposed architecture in figure 2 different from other popular communication methods in online learning that use attention architectures, such as TARMAC [1], ATOC [2], etc

3. in figure 2, why would the observations be passed through and FC layer and then combined again through another FC layer with the outputs of the attention module? why not, for instance, passing the observations just through one FC layer instead of 2 (or 3)?

4. do the agents communicate the same information both in the fully and partially obserable conditions?



[1] TarMAC: https://arxiv.org/abs/1810.11187

[2] Learning Attentional Communication for Multi-Agent Cooperation: https://arxiv.org/abs/1805.07733

[3] QPLEX: https://arxiv.org/abs/2008.01062

---

### Official Review · Reviewer_bAo5 · 2025-11-01

**Soundness:** 2
**Presentation:** 3
**Contribution:** 2
**Rating:** 4
**Confidence:** 4

**Summary:**

This paper addresses a key challenge in **offline multi-agent reinforcement learning (MARL)**: how to enable communication among agents even when **no communication actions are observed** in the offline dataset. The proposed method introduces a **learnable attention-based communication mechanism**, trained offline without access to explicit communication actions, but activated during online evaluation. Empirical results demonstrate consistent improvements over several baselines across standard offline MARL benchmarks.

**Strengths:**

1. The paper tackles an important and under-explored problem: **learning communication policies from purely offline datasets without communication supervision**, which is practically relevant in many real-world settings.
2. The authors propose a **simple but effective attention-based message passing mechanism**, and show it can enhance performance in offline MARL tasks when communication is enabled at evaluation time.

**Weaknesses:**

1. **Misleading novelty claim**
   The paper claims to be the first to explore learnable communication in offline MARL, but this is inaccurate. Prior works have already introduced **communication mechanisms in both online and offline settings**:

   * **MASIA** [1] proposed an **information extraction + aggregation module** that is highly analogous to the attention-based communication in this paper. MAICA can be seen as a **lightweight special case** of MASIA's architecture.
   * **Intention Sharing** [2] also adopts an attention-based message-sharing mechanism and reports **offline evaluation results** in cooperative scenarios.

2. **Methodological overlap and simplicity**
   While the paper introduces an attention-based approach, it lacks **technical novelty** over previous works. The proposed method reduces to a **lightweight attention model for inter-agent communication**, with no explicit theoretical insight or architecture-level innovation beyond existing paradigms.

3. **Inadequate comparison with communication baselines**
   The empirical evaluation does not include **other attention-based or learned communication approaches**, even though such methods (e.g., Intention Sharing, MASIA) already exist. A comparison would be necessary to fairly evaluate the contribution.

4. **Unrigorous writing and referencing**
   There are several problematic claims in the writing. For example:

   * Line 137–144 refers to two behavior-constrained methods (OMIGA and FACMAC+B3C) as if they exhaust the offline MARL constraint literature. However, other frameworks such as **IQL and CFCQL** are ignored, despite being earlier and more widely adopted.
   * Equation (1) defines `f(π, π)` without distinguishing between learning and behavior policies—this is unclear and mathematically imprecise.
   * The claim that the method extends prior behavior-constraint methods to the **multi-agent setting** should explicitly state **offline multi-agent**, as online MARL does not face the same constraints.

5. **Assumption–Performance contradiction**
   The paper suggests that communication learned offline (from passive data) can be activated at test time with strong performance gains. If this is generally true, then **large-scale distributed offline MARL with attention-based communication** should already perform well, yet existing literature does not support this trend.

**Questions:**

1. How is your method technically distinct from MASIA and Intention Sharing?
2. Why are communication baselines like DIAL, IC3Net, and attention-based intention sharing excluded from experiments?
3. Does your method assume **global observability**, or are the results still valid under partial observability (i.e., decentralized settings)?
4. Can the learned attention weights be interpreted as meaningful communication, or are they just implicit coordination priors?

---

### Official Review · Reviewer_hYsc · 2025-11-01

**Soundness:** 4
**Presentation:** 4
**Contribution:** 3
**Rating:** 8
**Confidence:** 4

**Summary:**

The paper proposes CODA, a framework that enables agents to learn communication policies from offline datasets generated by non-communicating agents. The core idea is to integrate an attentional communication network into existing offline MARL algorithms (OMIGA and FACMAC+B3C) and train it using gradient flows from the policy objectives. Although the method is simple, the authors demonstrate its the effectiveness through extensive empirical experiments across various scenarios, providing insights in how and why communication is helpful for multi-agent systems.

**Strengths:**

This paper studies a novel problem, offline communication learning from datasets without communications. The paper is well-motivated and easy to follow. The authors provide strong empirical validation.

**Weaknesses:**

It's a pity that this paper did not dive deep into the information of the messages, i.e. to communicate efficently by conveying the minimal amount of information.

**Questions:**

Can the authors provide an ablation study on the dimension of the message $d_k$?

---

### Note · Authors · 2025-12-08

I have read and agree with the venue's withdrawal policy on behalf of myself and my co-authors.